# Circulating Tumor DNA Detects Minimal Residual Disease in Patients with Locally Advanced Rectal Cancer After Total Neoadjuvant Therapy

**DOI:** 10.3390/cancers17152560

**Published:** 2025-08-03

**Authors:** Jin K. Kim, Ashley J. Alden, Sarah Knaus, Rishabh Thakkar, Lisa Moudgill, Allen Chudzinski, Paul Cavallaro, Carolina Martinez, Robert D. Bennett, Jorge Marcet

**Affiliations:** 1Department of Surgery, Division of Colorectal Surgery, University of South Florida Morsani College of Medicine, Tampa General Hospital, Tampa, FL 33601, USA; jinkim@usf.edu (J.K.K.); lisa.moudgill@va.gov (L.M.); chudzinski@usf.edu (A.C.);; 2Department of Surgery, Veterans Affairs, James A. Haley Veterans Hospital, Tampa, FL 33612, USA

**Keywords:** rectal cancer, watch and wait, organ preservation, liquid biopsy, ctDNA, total neoadjuvant therapy

## Abstract

Patients with locally advanced rectal cancer (LARC) who achieve a complete clinical response (cCR) after total neoadjuvant therapy may safely avoid radical surgery under a watch-and-wait protocol, but the clinical tools for assessing response have limitations. Unfortunately, nonoperative management of patients who harbor residual cancer will have negative implications on survival, and thus, accurate patient selection is critical. Circulating tumor DNA (ctDNA) testing has shown some promise in detecting minimal residual disease. This study aims to investigate ctDNA and its ability to detect minimal residual disease as an adjunct to reassess tumor response in locally advanced rectal cancer patients who received total neoadjuvant therapy.

## 1. Introduction

Colorectal cancer is the third most common cancer and is the second leading cause of cancer-related death worldwide [1]. The incidence of rectal cancer alone in the United States is estimated to result in 46,950 new diagnoses in 2025 [2]. Moreover, there has been an alarming rise in the incidence of young-onset rectal cancer, where it is estimated that a third of all new rectal cancer diagnoses will be in patients younger than 50 [3].

Total neoadjuvant therapy (TNT) with radiotherapy and systemic chemotherapy has become a standard treatment regimen for patients with locally advanced rectal cancer (LARC). Patients who demonstrate a complete or near-complete clinical response may be offered nonoperative management (NOM) to safely omit total mesorectal excision (TME). [4,5,6]. The OPRA trial, which incorporated TNT and NOM for patients with a complete or near-complete response, has shown that 50% of LARC patients were able to safely omit TME [6]. This has a tremendous impact on patients’ quality of life, as TME can have detrimental effects on sexual, urinary, and bowel function [7]. Despite the well-described protocols for assessing response after TNT, selection of patients who will benefit from NOM can be challenging. Clinical response assessment to TNT relies on careful evaluation, including a digital rectal exam, endoscopy, and pelvic MRI. Each of these exams has its own limitations, and the results can be discordant [8,9,10]. Approximately 30% of patients under NOM can develop local tumor regrowth while on surveillance, which would then require salvage surgery [11,12]. Although salvage surgery is successful in most patients with local regrowth, these patients have worse overall survival compared to patients who have a sustained complete response, which could be due to the progression of untreated minimal residual disease (MRD) [11,13]. Therefore, it is imperative to find other methods to more accurately detect MRD.

Liquid biopsy with ctDNA testing is being explored to detect MRD and metastases [14,15,16]. ctDNA has the potential to detect recurrence before clinical or radiological signs appear and can be a prognostic biomarker after curative surgery [15,17,18,19]. However, there is a paucity of data on the utility of ctDNA testing to detect MRD in patients with LARC after TNT. We hypothesized that ctDNA could detect MRD in LARC patients treated under a watch-and-wait protocol, and it could possibly be considered as a biomarker for identifying patients who may not benefit from NOM.

## 2. Materials and Methods

### 2.1. Patients

This is a single-institution retrospective case series study of LARC patients diagnosed from 2019 to 2023 who were treated with TNT and were managed with a watch-and-wait protocol at a quaternary care center with expert colorectal surgeons. TNT comprises radiotherapy in the form of short-course radiotherapy (25 Gy) or long-course chemoradiation (50.4–54 Gy with concurrent 5-flurouracil based chemotherapy), along with systemic chemotherapy intended for 8 cycles of FOLFOX or 6 cycles of CAPOX. Baseline tumor staging was performed per the National Comprehensive Cancer Network’s guidelines, with pelvic MRI, endoscopic and digital rectal exams, and chest/abdomen/pelvis CT [20]. In this study cohort, the treating colorectal surgeon ordered ctDNA testing as an adjunct to standard serial clinical response assessments.

### 2.2. Response Assessments

Response assessments consisted of pelvic MRI, CT chest/abdomen/pelvis, digital rectal exam, and flexible sigmoidoscopy 8–12 weeks after TNT. Serial surveillance consisted of a digital rectal exam and flexible sigmoidoscopy every 3–6 months and pelvic MRI every 6 months for the first 2 years [6,21,22]. Response assessment followed a three-tier schema of a clinical complete response (cCR), near-complete response (nCR), and incomplete response (iCR). cCR was defined as no residual tumor felt on digital rectal exam, presence of a white scar and/or telangiectasia during endoscopy, and no T2 intermediate signal and absence of restriction to diffusion on MRI. nCR was defined as a minor mucosal abnormality/irregularity felt during exams, superficial ulceration and/or mild erythema on endoscopy, and dense fibrosis with minimum T2 intermediate signal and minimal areas of restriction to diffusion on MRI. iCR was defined as a palpable tumor or visible tumor on endoscopy or MRI [10,21,22,23]. However, the language used to document responses in the medical charts during this study period was not uniform, as many of our patients obtained imaging at outside facilities that do not follow our institutional reporting structure. For the purposes of this study, we categorized response assessments as suspicious for tumor, indeterminate, or negative for tumor after best matching the findings with previously published assessment guidelines [9,10,21,24].

### 2.3. Local Regrowth

Local regrowth was defined as radiologic or clinical reappearance of tumor during surveillance in patients with a cCR or nCR on the watch-and-wait protocol. Patients with local regrowth were offered salvage surgery either as a local excision (LE) or TME.

### 2.4. ctDNA Testing

Tumor-informed ctDNA testing was performed with Signatera (Natera Inc., San Carlos, CA, USA). Cell-free DNA was extracted from patient plasma to detect ctDNA. Formalin-fixed paraffin-embedded tissue slides of pretreatment rectal cancer biopsies with matched patient whole-blood samples collected in Streck tubes (Cell-Free DNA BCT; Streck, La Vista, NE, USA) were sent for next-generation sequencing to infer up to 16 patient- and tumor-specific somatic single-nucleotide variants in plasma cell-free DNA using a multiplex PCR-based approach. Plasma samples with at least two tumor-specific variants detected above Nateria Inc’s pipeline threshold were defined as ctDNA-positive. This assay has been reported to have greater than 95% sensitivity at 0.01% mean variant allele frequency with 99.7% specificity [19]. The average turnaround time for the initial assay design was 3–4 weeks and approximately 1 week for subsequent samples.

### 2.5. Statistical Analysis

Statistical analysis was performed with GraphPad Prism version 10.4.1. A Mann–Whitney U test was performed for continuous variables, and Fisher’s exact test was performed for categorical variables. Log-rank test was performed to compare median follow-up.

## 3. Results

### 3.1. Patient Groups by ctDNA Positivity

A total of 28 LARC patients had ctDNA testing as part of their response assessment after TNT. In total, 9 patients had positive ctDNA, and 19 had negative ctDNA during follow-up after therapy. Baseline characteristics of these two groups are presented in Table 1.

There was no difference in mean age (60 versus 66, *p* = 0.25), sex (*p* > 0.9), or mean tumor distance from the anal verge (5.7 cm versus 5.9 cm, *p* = 0.84) in the ctDNA-positive versus ctDNA-negative cohorts, respectively. Clinical T3 tumors were predominant in both groups. The majority of patients had clinically positive lymph node metastases in both groups (78% versus 68%, *p* > 0.9). There was no significant difference in the distribution of clinical T and N stage tumors in the two groups. The vast majority of tumors had proficient mismatch repair (MMR) expression.

The median follow-up time after completion of TNT was similar in both the ctDNA-positive versus the ctDNA-negative groups (1.81 versus 1.47 years, *p* = 0.72). Definitive surgical management for the rectal primary was categorized as TME or local excision (LE), and the remainder were categorized as NOM. In the ctDNA-positive group, six patients developed local regrowth. Two of these underwent LE, and four underwent TME. All six of these ctDNA-positive patients had residual cancer in the specimen. In the ctDNA-negative group, four patients developed local regrowth, and all four patients underwent TME. In this study, one out of the four TME patients had a pathological complete response. There was a significantly higher rate of NOM in the ctDNA-negative group versus the ctDNA-positive group (74% vs. 33%, *p* = 0.035). Two patients (22%) in the ctDNA-positive group and one patient (5%) in the ctDNA-negative group developed distant metastases. No patients had local recurrence after surgery in this study cohort.

To better visualize the utility of ctDNA testing in detecting minimal residual disease in the rectum after TNT, we created an overview plot of the patients grouped by ctDNA positivity (Figure 1).

We aligned time zero as the last day of TNT and plotted the ctDNA test results over time for each patient. Our usual means of clinical restaging, including pelvic MRI and endoscopic/digital rectal exam, were also plotted with the clinical interpretation suspicious for tumor, negative for tumor, or indeterminate. Any surgical intervention in the rectum (LE and TME) was also plotted as an event to understand if the ctDNA test correlated with MRD in the rectum.

### 3.2. ctDNA Can Potentially Detect MRD in the Rectum in Patients Under Watch-and-Wait Protocol After TNT

In the ctDNA-positive group, four out of six patients who required surgery for suspected local regrowth had positive ctDNA test results prior to surgery without evidence of distant metastases (Figure 1).

Patient LARC20 had MRI and endoscopic exams, which were suspicious for residual disease at initial response assessment after completing TNT. ctDNA was negative initially but turned positive on two consecutive draws. Repeat endoscopic examination around this time was still suggestive of residual rectal cancer. The patient underwent TME and was found to have ypT3N0 rectal cancer. ctDNA was cleared postoperatively.

Patient LARC27 had a similar case, but the MRI scan was read as negative for possible residual cancer. However, on serial endoscopic exams, the patient appeared to have worsening features suggestive of residual cancer. ctDNA was positive in the pre-resection phase. The patient underwent a TME and was found to have ypT3N0 rectal cancer. Subsequent postoperative ctDNA draws were negative.

Patient LARC28 was on the watch-and-wait protocol for almost a year without significant findings. ctDNA was negative initially but turned positive. On surveillance MRI, concerning features in the inguinal nodes were found, but they were indeterminate for residual rectal cancer. Endoscopic exams did not detect any disease. The patient underwent a series of LE and inguinal node dissection and was found to have ypT2Nx with 2/7 positive inguinal lymph nodes.

Patient LARC26 continued on the watch-and-wait protocol after local excision, which came back as tubulovillous adenoma. A little over a year out from TNT, the patient had positive ctDNA on serial draws. MRI and endoscopic exams were also suspicious for residual rectal cancer. This patient also underwent TME, revealing ypT3N0 rectal cancer.

Patient LARC4 did not have ctDNA drawn before TME. The postoperative ctDNA test was initially negative but became positive on serial draws. The clinical significance of this is yet unclear, as we have not yet detected obvious locoregional recurrence or clinical metastases.

Patient LARC10 had a negative ctDNA at the time of LE (ypT2Nx) and tested positive on postoperative surveillance draws. We have not yet detected obvious clinical metastases. This patient continues to be on close surveillance to determine whether this is a false-positive test.

Patients LARC2 and LARC23 also developed positive ctDNA on watch and wait without definite evidence for local regrowth or distant metastases.

In the ctDNA negative cohort, four patients required surgery for suspected local regrowth (Figure 1). LARC 14 had a negative ctDNA preoperatively, with indeterminate results on MRI and initial endoscopy. After a repeat endoscopic exam, which had findings suspicious for tumors, this patient underwent TME but had a pathological complete response.

Patients LARC24, LARC11, and LARC9 all had negative preoperative ctDNA results but had suspicious findings on endoscopy, so they underwent TME with residual tumor in the specimen.

Patients LARC7, LARC3, LARC8, LARC12, LARC19, and LARC16 all initially had endoscopic or imaging findings consistent with a near-complete response, which evolved to a complete clinical response over time. Patients LARC25, LARC5, LARC6, LARC13, LARC15, LARC17, LARC18, LARC21, and LARC1 had sustained a complete clinical response since the initial response assessment. These patients had negative ctDNA on all surveillance draws.

### 3.3. Potential Role of ctDNA in Detecting Distant Metastases

Three patients in our study developed distant metastases: two with pulmonary metastases and one with liver metastasis. We created another overview plot of these patients, including CT chest/abdomen/pelvis results and ctDNA testing, to visualize whether ctDNA testing can be a helpful marker for detecting metastases (Figure 2).

Interestingly, ctDNA turned positive in two patients weeks before CT detected metastases. Patient LARC13 did not have positive ctDNA prior to diagnosis on CT, but the interval from the ctDNA draw to the positive finding on CT was almost 2 years.

## 4. Discussion

Patients with a complete response to TNT may benefit from NOM, but the clinical tools for assessing response have limitations. Unfortunately, the NOM of patients with residual cancer can have negative implications on survival as they are more likely to develop distant metastases [11,13]. Implementation of a molecular test, such as ctDNA, to detect MRD is therefore attractive. In this case series of 28 patients, we showed the potential of using ctDNA as a biomarker for detecting MRD in the rectum after TNT. Of the nine patients who had residual cancer in the surgical specimen, four (44%) had positive ctDNA results prior to surgery. Sustained ctDNA negativity was associated with organ preservation in this study. Furthermore, we demonstrated the potential of ctDNA as a biomarker for detecting distant metastases during surveillance.

There are several limitations to our study. This is a small retrospective case series with potential selection bias for patients, as ctDNA testing was not standardized. Of the 28 patients, 1 patient had an MMR-deficient tumor, and three patients had an unknown MMR status. Given the small sample size, we elected to group these samples for analyses, but it is important to note that response to TNT is different by MMR status and that tumor DNA shedding is known to be higher in MMR-deficient tumors, which alters the sensitivity of the ctDNA assay [25]. Locally advanced rectal cancer patients with MMR deficiency are now treated with neoadjuvant dostarlimab, a PD-1 inhibitor, which has been shown to achieve a 100% response rate [26]. Only patients with sufficient tumor biopsy tissue to run a tumor-informed ctDNA assay were analyzed in this study. While it is beyond the scope of this study, tumor-agnostic (plasma-only) ctDNA assays could also be considered in patients with insufficient tumor tissue. Additionally, while we established a standardized institutional protocol for clinical response assessment, the language used to document clinical response was variable in the medical charts during this study period. While on surveillance, four patients turned positive with respect to ctDNA, without clinical signs of recurrence. It is unclear whether these are false-positive tests or truly indicative of MRD, as the Signatera ctDNA assay has been shown to have a 8.7-month lead time in detecting recurrence in colon cancer [27]. A longer follow-up interval may be needed to make conclusions in this subpopulation. In this study, we did not alter treatment based on ctDNA results and will rely on larger prospective trials to elucidate whether treating ctDNA positivity results in improved outcomes.

Small studies have highlighted the association of ctDNA with clinical response to neoadjuvant chemoradiation [28,29,30]. In a study of 44 LARC patients, ctDNA had a sensitivity of 23% and a negative predictive value of 47% in detecting residual disease after TNT [31]. While we did not obtain pretreatment ctDNA in our patients, clearance of baseline ctDNA positivity after neoadjuvant therapy has also been shown to be associated with response [32]. For our study, we stratified patients by ctDNA positivity, but Signatera also provides quantitative measurements of tumor-derived mutant/mL of plasma and the variant allele fraction to infer the concentration of circulating tumor DNA in the plasma. Both quantitative measurements are correlated with each other and have been shown to be associated with tumor burden and overall survival in some studies [33]. The quantitative amount of tumor-derived mutant molecules/mL of plasma has shown correlations with the likelihood of recurrence in locally advanced colon cancer after resection in a recently reported interim analysis of the DYNAMIC-III trial [34]. Taken together, ctDNA testing offers a molecular-based biomarker that could help guide treatment in the future. ctDNA testing alone may not be sufficient to detect MRD, but it may provide value as a diagnostic adjunct in identifying the subset of patients with residual cancer who may not benefit from NOM.

The National Cancer Institute Taskforce on colon and rectal–anal cancers recently published recommendations based on expert consensus to help standardize the timeline of ctDNA collection. The Taskforce recommends ctDNA collection at least 4 weeks after the completion of neoadjuvant therapy and within a 2-week window of clinical restaging [16]. Several clinical trials are currently recruiting patients to study the role of ctDNA as a tool for response assessment in LARC after TNT. The Janus trial is a multicenter phase II trial studying two arms of TNT with long-course chemoradiation followed by consolidation FOLFOX versus FOLFIRINOX chemotherapy (NCT05610163) [35]. ctDNA is being collected during TNT and while on surveillance to correlate with response and outcomes. A single-institution observational trial is collecting ctDNA at pretreatment, during TNT, and while on surveillance (NCT05081024). These trials will hopefully shed more light on the role of ctDNA in detecting MRD after TNT and provide an improved program for nonoperative management.

## 5. Conclusions

Liquid biopsies represent an intriguing adjunct for identifying patients with MRD after TNT for rectal cancer. Incorporation of ctDNA should be considered in surveillance protocols or response assessment of LARC patients who desire organ preservation.

## Figures and Tables

**Figure 1 cancers-17-02560-f001:**
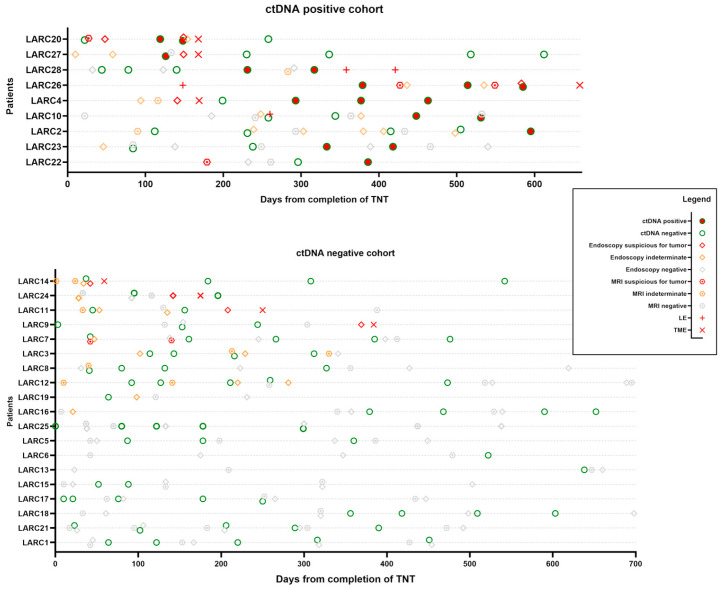
Overview plot of all patients with positive ctDNA during follow-up. Events are arranged from the last date of total neoadjuvant therapy (TNT) for each patient. Chronological view of the results of clinical response assessments, including pelvis MRI, digital rectal exam, endoscopic exam (results grouped together as endoscopy in this figure), and ctDNA, is displayed. Locally advanced rectal cancer (LARC) patients who required rectal surgery by either local excision (LE) or total mesorectal excision (TME) are listed at the top.

**Figure 2 cancers-17-02560-f002:**
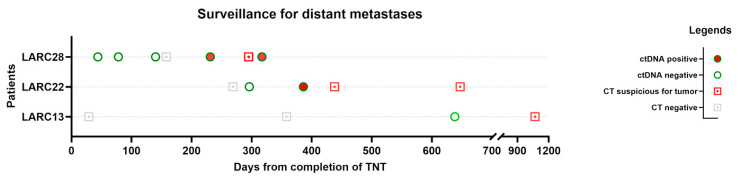
Overview plot of patients with distant recurrence. Events are arranged from the last date of total neoadjuvant therapy (TNT) for each patient. Chronological view of the results of chest/abdomen/pelvis CT and ctDNA is displayed.

**Table 1 cancers-17-02560-t001:** Tumor characteristics and outcomes of patients by ctDNA positivity after total neoadjuvant therapy.

	ctDNA-Positive	ctDNA-Negative	*p*-Value
Number of patients	9	19	
Age (years) (mean, std dev)	60 (11)	66 (12)	0.25
Sex (N, %)			>0.9
Male	5 (56%)	12 (63%)	
Female	4 (44%)	7 (37%)	
cT (N, %)			0.21
2	1 (11%)	1 (5%)	
3	6 (67%)	17 (90%)	
4	2 (22%)	1 (5%)	
cN (N, %)			0.65
0	2 (22%)	6 (32%)	
1	6 (67%)	9 (47%)	
2	1 (11%)	4 (21%)	
MMR status (N, %)			>0.9
MMR-proficient	8 (89%)	16 (84%)	
MMR-deficient	0 (0%)	1 (5%)	
Unknown	1 (11%)	2 (11%)	
Tumor distance (cm) (mean, std)	5.7 (3.8)	5.9 (4.2)	0.84
Total neoadjuvant therapy (N, %)	9 (100%)	19 (100%)	>0.9
Local Regrowth (N, %)	6 (66%)	4 (21%)	0.035
Definitive treatment of primary tumor (N, %)			0.041
LE	2 (22%)	0 (0%)	
TME	4 (44%)	4 (21%)	
NOM	3 (33%)	14 (74%)	
Median follow-up (years)	1.81	1.47	0.72
Distant recurrence (N, %)	2 (22%)	1 (5%)	0.23
Local recurrence (N, %)	0 (0%)	0 (0%)	
Resection Pathology (N)	6	4	
pCR	0 (0%)	1 (25%)	
Residual tumor	6 (100%)	3 (75%)	

MMR = Mismatch repair; LE = local excision; TME = total mesorectal excision; NOM = nonoperative management; pCR = pathological complete response. *p* = 0.035 (surgery vs. NOM).

## Data Availability

Data are contained within the article.

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
