# Peer review of "Circulating Tumor DNA Detects Minimal Residual Disease in Patients with Locally Advanced Rectal Cancer After Total Neoadjuvant Therapy"

_cancers, 2025, doi:10.3390/cancers17152560_

Round 1
Reviewer 1 Report
Comments and Suggestions for Authors
In the submitted manuscript, Kim and colleagues set out to investigate whether ctDNA could be leveraged to detect minimal residual disease in patients with locally advanced rectal cancer (LARC). Currently, the selection of LARC patients for nonoperative management following total neoadjuvant therapy (TNT) is clinically challenging, and assessing the efficacy of TNT has limitations and often returns discordant results. Thus, the development of ctDNA as biomarker to evaluate the efficacy of total neoadjuvant therapy could be of significant impact to the management of LARC, as the detection of ctDNA could be used as powerful tool to aid in identifying patients appropriate for de-escalation from total mesorectal excision following TNT.
In this single institution retrospective study testing for ctDNA presence in 28 patients with LARC following TNT, Kim et al positively detected ctDNA in 9 patients. Of the 9 patients with positive ctDNA, 67% (6/9) required surgery to remove residual rectal cancer, while only 21% (4/19) of patients with undetectable ctDNA eventually required surgery. Based on these findings, the authors conclude that testing for the presence ctDNA following TNT has the potential to detect minimal residual disease in LARC.
The data reported by Kim et al are clearly presented, of clinical significance, and are suitable for publication only after careful consideration of the following critiques:
CRITIQUES
- The most prominent limitation of this study, as identified by the authors, is the small sample size of appropriate LARC patients tested (n =28). Without immediately spending time and resoruces by adding more patients to this pilot study, the authors could offset this limitation by using specific statistics defining the percent of LARC patients that could potentially be responsive to TNT and benefit from a ctDNA biomarker that would qualify for non-operative management. Describing the magnitude of the problem will help with the over impact of the findings.
- The authors have defined the manner in which the detection of ctDNA in LARC could improve patient management, but how would further characterization of detectable ctDNA be additionally beneficial? There are several studies in multiple disease settings that describe the identification of genetic and genomic aberrations in ctDNA that have diagnostic, prognostic and therapeutically predictive potential. Sequencing of detectable ctDNA from LARC patients in a study such as this one could lead to 1) predicting response to TNT using baseline samples, and 2) provide rationale to discover why LARC patients with detectable ctDNA don’t progress to the need for surgery (i.e. the 3/9 patients that didn’t require excision in this study). Even if the authors don’t have immediate resources or available material to sequence the LARC patients in this study, they should consider mentioning the impact of more in-depth characterization of ctDNA in the Discussion section of the manuscript.
Author Response
We would like to thank the reviewer for providing thoughtful suggestions to help improve our manuscript! We have revised the manuscript to incorporate the suggestions. Please see below our point by point responses to the reviewer's comments:
Comment 1: The most prominent limitation of this study, as identified by the authors, is the small sample size of appropriate LARC patients tested (n =28). Without immediately spending time and resources by adding more patients to this pilot study, the authors could offset this limitation by using specific statistics defining the percent of LARC patients that could potentially be responsive to TNT and benefit from a ctDNA biomarker that would qualify for non-operative management. Describing the magnitude of the problem will help with the over impact of the findings.
Response to comment 1: This is an important suggestion from the reviewer. We have added a new first paragraph in the Introduction to better describe the incidence of rectal cancer and the magnitude of the problem for the readers.
Comment 2: The authors have defined the manner in which the detection of ctDNA in LARC could improve patient management, but how would further characterization of detectable ctDNA be additionally beneficial? There are several studies in multiple disease settings that describe the identification of genetic and genomic aberrations in ctDNA that have diagnostic, prognostic and therapeutically predictive potential. Sequencing of detectable ctDNA from LARC patients in a study such as this one could lead to 1) predicting response to TNT using baseline samples, and 2) provide rationale to discover why LARC patients with detectable ctDNA don’t progress to the need for surgery (i.e. the 3/9 patients that didn’t require excision in this study). Even if the authors don’t have immediate resources or available material to sequence the LARC patients in this study, they should consider mentioning the impact of more in-depth characterization of ctDNA in the Discussion section of the manuscript.
Response to comment 2: Thank you for this thoughtful suggestion. We are excited about the future of personalized medicine that we hope will come to fruition with improved molecular assays.
In a landmark paper of genomic and transcriptomic predictors of response in rectal cancer, pretreatment tumor genomics has not shown to be associated with tumor response (PMID: 35970919). While our current study is small and exploratory in nature, we were early adopters of ctDNA testing when it first came to market in 2019 because we saw the potential of ctDNA testing to add to our restaging algorithm for locally advanced rectal cancer patients. In the future, we plan to incorporate a more structured timeline to better align the data in subsequent patients.
For the reviewer’s second comment, we are also very curious about the 4 patients with detectable ctDNA who did not have signs of recurrence within the study time period. We postulate that these are either false positive tests or that these patients will ultimately develop signs of local or distant recurrence later with more follow up at ctDNA has been shown to have a lead time of 8.7 months in detection of recurrence in colon cancer (PMID: 31070691). We have revised the Discussion to talk through our thought process. An interim analysis of the CIRCULATE-Japan GALAXY trial observational study in colon cancer shed light on strong association of ctDNA positivity with minimal residual disease (PMID: 39284954). In their observational study, patients were determined to be ctDNA “positive” or “negative” 1 month after curative resection. ctDNA positivity after resection was the strongest marker for survival. Patients who were ctDNA positive after resection also appeared to benefit from adjuvant chemotherapy. Sustained clearance of ctDNA was associated with improved outcomes compared to transient clearance of ctDNA. Based on these findings, we are also curious whether more in-depth characterization of ctDNA could hold additional valuable information.
Reviewer 2 Report
Comments and Suggestions for Authors
Dr. Jin and colleagues present a small, retrospective, single-institution study investigating whether circulating tumor DNA (ctDNA) can help guide nonoperative management (NOM) in patients with locally advanced rectal cancer (LARC) who achieve a clinical complete response (cCR) after total neoadjuvant therapy (TNT). The study includes 28 patients and suggests ctDNA may support decision-making in this setting.
While the research question is of clinical interest, the study’s methodological limitations significantly weaken the validity and interpretability of the results.
Major Comments
- ctDNA Testing Issues
- The manuscript does not clearly define when ctDNA was measured. The phrase “during follow-up” is vague, and Figure 1 suggests testing occurred at multiple, inconsistent timepoints. This heterogeneity limits the ability to interpret ctDNA’s predictive value, especially for early identification of complete responders—the primary clinical challenge the study aims to address.
- The manuscript omits critical assay parameters, including the number and type of mutations tracked, panel size, detection limit, frequency of testing, and turnaround time. These are essential to assess assay reliability, sensitivity, and potential clinical application.
- The use of a tumor-informed assay (Signatera) requires sufficient tumor tissue, which is often limited in rectal cancer biopsies. This limitation is not acknowledged. Plasma-only assays should be briefly discussed as a more feasible alternative for real-world application.
- Non-standardized cCR Assessment
The criteria for evaluating clinical response are not well defined and appear to vary across patients. Timing of imaging post-TNT is unclear, and interpretation was not standardized. Retrospective application of MRI tumor regression grade (e.g., mrTRG) could help, but would not fully address this methodological gap. - Unclear Definition of Local Regrowth
Among ctDNA-negative patients, 4 experienced local regrowth, yet one had a pathological complete response at surgery, suggesting a false-positive clinical assessment. The manuscript should clarify how regrowth was defined (e.g., imaging, biopsy, endoscopy) and discuss the implications for overtreatment. - Pooling of MSI and MSS Patients
Combining MSS and MSI-H patients without adjustment is problematic, given their distinct biology, treatment sensitivity, and ctDNA shedding patterns. While MSI-H patients were few (n=1), this pooling introduces potential confounding and should at least be discussed. - Methodological Weaknesses
- No prespecified hypothesis or statistical analysis plan is reported. The study appears exploratory and at risk of data dredging.
- The small sample size (n=28) and group imbalance limit statistical power. For instance, a higher proportion of ctDNA+ patients were cT4 (22% vs. 5%), but the absolute numbers (2 vs. 1) preclude meaningful inference.
- No performance metrics (e.g., sensitivity, specificity, AUC, NPV, PPV) are provided, despite evaluating ctDNA as a potential diagnostic biomarker. This significantly limits the translational impact of the findings.
- Potential False Positives
Several patients became ctDNA-positive during surveillance without subsequent clinical or radiologic evidence of recurrence. While the authors suggest longer follow-up is needed, this raises important questions about the assay’s specificity and risk of overdiagnosis, which deserve more in-depth discussion. - Short Follow-Up
The median follow-up of 1.5–1.8 years may not be sufficient to detect late local or distant recurrences, particularly given the natural history of LARC. Any conclusions regarding long-term safety or utility of ctDNA in surveillance are premature. - Overgeneralized Claims and Weak Contextualization
The abstract and discussion use vague terms like “respond well” and “attractive treatment option,” which are not aligned with current guidelines. - NOM should only be considered for patients with confirmed cCR and in high-volume expert centers.
- Existing tools to assess cCR, such as the MSKCC algorithm (PMID: 35512720), are not cited and should be acknowledged.
- The characterization of TNT as “chemoradiotherapy” omits short-course regimens (e.g., RAPIDO), and the claim that NOM is “safe” is overly broad given ongoing uncertainty about long-term distant relapse. The only prospective study with distant relapse-free survival (DRFS) as a primary endpoint—NO-CUT—was presented at ESMO 2024 but is not yet published.
Author Response
First off, we would like to thank our reviewer for providing us with critical and thoughtful feedback and suggestions. I've learned more about my own project during this revision process and am deeply grateful for your time. We have revised the manuscript accordingly. Please see below our point by point response to reviewer's comments.
Comment 1: The manuscript does not clearly define when ctDNA was measured. The phrase “during follow-up” is vague, and Figure 1 suggests testing occurred at multiple, inconsistent timepoints. This heterogeneity limits the ability to interpret ctDNA’s predictive value, especially for early identification of complete responders—the primary clinical challenge the study aims to address.
Response to comment 1: Thank you to the reviewer for these salient points. We agree with the reviewer that these data are heterogeneous and is a limitation of our study. We already discuss this as a limitation of our study in the discussion. While our study is small and exploratory in nature, we were early adopters of Signatera ctDNA testing when it first came to market in 2019 because we saw the potential of ctDNA testing to add to our restaging algorithm in locally advanced rectal cancer patients. And as we discuss in the manuscript, we did not deviate from our clinical practice based on the ctDNA information. This was purely an observational case series study to evaluate whether ctDNA testing can detect disease in patients after total neoadjuvant therapy. In the future, we will be incorporating a more structured ctDNA timeline to better align the data in future analyses.
Comment 2: The manuscript omits critical assay parameters, including the number and type of mutations tracked, panel size, detection limit, frequency of testing, and turnaround time. These are essential to assess assay reliability, sensitivity, and potential clinical application.
Response to comment 2: Thank you for this suggestion. We have revised the Method section to describe more in detail of how blood was collected and processed. We also included more technical details of Signatera ctDNA assay and added critical assay parameters, number of mutations tracked, detection limit and turnaround time. The frequency of testing was not standardized as we had noted in our manuscript.
Comment 3: The use of a tumor-informed assay (Signatera) requires sufficient tumor tissue, which is often limited in rectal cancer biopsies. This limitation is not acknowledged. Plasma-only assays should be briefly discussed as a more feasible alternative for real-world application.
Response to comment 3: This is a real logistical limitation of tumor-informed assays as suggested by the reviewer. However, we don’t think applying Signatera is considered a limitation for the sake of this study. Tumor informed assays generally offer a higher sensitivity of detection compared to tumor-agnostic assays (plasma-only) and in our opinion, is a better test to detect MRD. Nonetheless, we have added a sentence in the Discussions to mention plasma-only assays as a potential alternative when tumor tissues are insufficient.
Comment 4: The criteria for evaluating clinical response are not well defined and appear to vary across patients. Timing of imaging post-TNT is unclear, and interpretation was not standardized. Retrospective application of MRI tumor regression grade (e.g., mrTRG) could help, but would not fully address this methodological gap.
Response to comment 4: We apologize if the evaluation of clinical response was not well described in the initial version of the manuscript. We are a group of expert colorectal surgeons at a quaternary care institution that sees the highest volume of rectal cancer patients in the region. We are part of the Rectal Cancer Consortium and have been an active clinical site for several clinical trials including the more recent OPRA trial (PMID: 26474521, PMID: 26187751, PMID: 35483010). We have been assessing response as described in the OPRA trial. We have revised the Methods portion to clearly declare our response assessment protocol. However, these are real-world data and shows the real life variation in surveillance interval and heterogeneity in language of documenting response in the medical charts. As we are a quaternary referral center, many of the patients that we see come to us with MRI imaging and reports that were performed at outside facilities which often do not have synoptic reporting. While we reassess the imaging with our own radiologists at multidisciplinary tumor board to determine radiologic response and formulate a treatment decision, the level of detail in clinical documentation of complete, near-complete, or incomplete response was lacking as these patients were not part of a clinical trial which require more robust documentation. Our study was retrospective in nature and data collected from chart review.
Comment 5: Among ctDNA-negative patients, 4 experienced local regrowth, yet one had a pathological complete response at surgery, suggesting a false-positive clinical assessment. The manuscript should clarify how regrowth was defined (e.g., imaging, biopsy, endoscopy) and discuss the implications for overtreatment.
Response to comment 5: We have revised the Methods portion of the manuscript to better define how Local Regrowth was clinically identified. As better stated in our revised Introduction, clinical response assessment is imperfect and has implications for overtreatment and undertreatment.
Comment 6: Combining MSS and MSI-H patients without adjustment is problematic, given their distinct biology, treatment sensitivity, and ctDNA shedding patterns. While MSI-H patients were few (n=1), this pooling introduces potential confounding and should at least be discussed.
Response to comment 6: Thank you for this important point. We agree with the reviewer that tumor biology, treatment, and ctDNA shedding patterns are different in MSS and MSI tumors. Given the already sample size, we did not stratify the patients by MMR status. We added a sentence in the Limitations paragraph of our Discussion to disclose this detail as suggested by the reviewer.
Comment 7: No prespecified hypothesis or statistical analysis plan is reported. The study appears exploratory and at risk of data dredging.
Response to comment 7: As we replied to reviewer’s comment #1, while our study is small and exploratory in nature, we were early adopters of Signatera ctDNA testing when it first came to market in 2019 because we saw the potential of ctDNA testing to add to our restaging algorithm in locally advanced rectal cancer patients. In the last paragraph of Introduction, we wrote our hypothesis: that ctDNA could detect MRD in LARC patients treated under a watch-and-wait protocol and possibly be considered as a biomarker to identify patients who may not be good candidates for NOM. This was purely an observational case series study to evaluate whether ctDNA testing can detect disease in patients after total neoadjuvant therapy. After reviewing our data and other groups’ published work, we will be incorporating a more structured ctDNA timeline to better align the data in future analyses.
Comment 8: The small sample size (n=28) and group imbalance limit statistical power. For instance, a higher proportion of ctDNA+ patients were cT4 (22% vs. 5%), but the absolute numbers (2 vs. 1) preclude meaningful inference.
Response to comment 8: This is a single institutional exploratory observational study to assess whether ctDNA testing can detect minimal residual in locally advanced rectal cancer patients after total neoadjuvant therapy who are being considered for possible watch and wait. Detection of ctDNA has several caveats including tumor burden and tumor DNA shedding. ctDNA has shown promise in colon cancer as a biomarker for recurrence after curative resection. While ctDNA is a promising tool to detect MRD, we have no robust data on its sensitivity and specificity in locally advanced rectal cancer after total neoadjuvant therapy which shrinks the tumor burden and alters the tumor microenvironment and vascular supply with radiation. Our main goal with our study was to assess for any positive signal to assess whether Signatera ctDNA testing is a feasible tool to aid response assessment after total neoadjuvant therapy in rectal cancer patients.
Comment 9: No performance metrics (e.g., sensitivity, specificity, AUC, NPV, PPV) are provided, despite evaluating ctDNA as a potential diagnostic biomarker. This significantly limits the translational impact of the findings.
Comment 9: As the reviewer suggests, performance metrics of sensitivity, specificity, AUC, NPV, PPV are important to consider for a biomarker. However, to calculate these statistics, the definition of a “true & false positive” is imperative. A well-designed study to evaluate this would require a larger number of patients and tissue specimen from all patients to define the “true positive” as cancer in the specimen. Given the small number of patients and few patients who undergo resection in our cohort, we do not think our data is positioned to answer this question. However, we do cite a small paper from a group who calculated these values on a cohort of 44 locally advanced rectal cancer patients of which 25 who had total neoadjuvant therapy followed by resection in the Discussion.
Comment 10: Several patients became ctDNA-positive during surveillance without subsequent clinical or radiologic evidence of recurrence. While the authors suggest longer follow-up is needed, this raises important questions about the assay’s specificity and risk of overdiagnosis, which deserve more in-depth discussion.
Response to comment 10: Reviewer #1 also raised the same question. We are also very curious about the 4 patients with detectable ctDNA who did not have signs of recurrence within the study time period. We postulate that either these are false positive tests or that these patients will ultimately develop signs of local or distant recurrence later with more follow up. An interim analysis of the CIRCULATE-Japan GALAXY trial observational study in colon cancer shed light on strong association of ctDNA positivity with minimal residual disease (PMID: 39284954). In their observational study, patients were determined to be ctDNA “positive” or “negative” 1 month after curative resection. ctDNA positivity after resection was the strongest marker for survival. Patients who were ctDNA positive after resection also appeared to benefit from adjuvant chemotherapy. Sustained clearance of ctDNA was associated with improved outcomes compared to transient clearance of ctDNA. Based on these findings, we certainly also believe more in-depth characterization of ctDNA could hold additional valuable information. The NCI Taskforce on Colon and Rectal-anal cancers has published expert consensus on ctDNA in colorectal rectal cancer and we added a paragraph in the Discussion section in our revised manuscript. There clearly is a false positive rate to any test but no robust data on the false positive rate of ctDNA to date. We hope that larger clinical trials will shed light on this for rectal cancer patients in the future.
Comment 11: The median follow-up of 1.5–1.8 years may not be sufficient to detect late local or distant recurrences, particularly given the natural history of LARC. Any conclusions regarding long-term safety or utility of ctDNA in surveillance are premature.
Response to comment 11: Thank you for this important point. Signatera ctDNA testing recently came into market and thus follow up length is short. We agree with the reviewer’s point and do not suggest any conclusions on the utility of ctDNA for surveillance to alter treatment decisions.
Comment 12: The abstract and discussion use vague terms like “respond well” and “attractive treatment option,” which are not aligned with current guidelines.
Response to comment 12: We have revised the wording of the abstract to be more clear and consistent with guidelines.
Comment 13: NOM should only be considered for patients with confirmed cCR and in high-volume expert centers.
Response to comment 13: As in our response to Reviewer’s comment #2, we are a group of expert colorectal surgeons at a quaternary care institution that sees the highest volume of rectal cancer patients in the region. We are part of the Rectal Cancer Consortium and have been an active clinical site for several trials including the more recent OPRA trial (PMID: 26474521, PMID: 26187751, PMID: 35483010). We assess response as described in the OPRA trial and provide watch and wait to patients with cCR or nCR.
Comment 14: Existing tools to assess cCR, such as the MSKCC algorithm (PMID: 35512720), are not cited and should be acknowledged.
Response to comment 14: Thank you for this excellent suggestion! We have added this citation within our Methods section for clinical response assessment. We also revised the text for our response assessment to more clearly define our protocol.
Comment 15: The characterization of TNT as “chemoradiotherapy” omits short-course regimens (e.g., RAPIDO), and the claim that NOM is “safe” is overly broad given ongoing uncertainty about long-term distant relapse. The only prospective study with distant relapse-free survival (DRFS) as a primary endpoint—NO-CUT—was presented at ESMO 2024 but is not yet published.
Response to comment 15: Thank you for this suggestion. In the revised manuscript, we better defined what TNT consisted of in these patients (which was either SCRT or LCRT plus FOLFOX or CAPOX). Of the 9 patients who were ctDNA positive, 2 patients received SCRT and 7 received LCRT. Of the 19 patients who were ctDNA negative, 6 patients received SCRT and 13 received LCRT. There was no obvious difference between the groups so we elected not to break down the numbers. There certainly is debate for SCRT versus LCRT but we feel that this is beyond the scope of our hypothesis which was to test if Signatera could even pick up a signal in these patients after being treated with TNT.
Round 2
Reviewer 2 Report
Comments and Suggestions for Authors
The authors have provided quite thoughtful and comprehensive responses. The authors have addressed most of my concerns and clarified key methodological limitations. I appreciate the added details on assay parameters, clinical response assessment, and follow-up. While some issues remain partially resolved—such as th lack of standardized response metrics, and absence of biomarker performance data—I recognize the exploratory nature of the study and the challenges posed by its retrospective design.